# Dynamics of competing SARS-CoV-2 variants during the Omicron epidemic in England

Oliver Eales [1,2] ✉, Leonardo de Oliveira Martins [3], Andrew J. Page [3], Haowei Wang[1,2], Barbara Bodinier[1,4], David Tang[1,4], David Haw[1,2], Jakob Jonnerby[1,2], Christina Atchison [1], Deborah Ashby [1], Wendy Barclay [5], Graham Taylor [5], Graham Cooke[5,6,7], Helen Ward [1,6,7], Ara Darzi [6,7,8], Steven Riley[1,2], Paul Elliott [1,2,6,7,9,10,12] ✉, Christl A. Donnelly [1,2,11,12] ✉ & Marc Chadeau-Hyam [1,4,12] ✉

The SARS-CoV-2 pandemic has been characterised by the regular emergence of genomic variants. With natural and vaccine-induced population immunity at high levels, evolutionary pressure favours variants better able to evade SARS-CoV-2 neutralising antibodies. The Omicron variant (first detected in November 2021) exhibited a high degree of immune evasion, leading to increased infection rates worldwide. However, estimates of the magnitude of this Omicron wave have often relied on routine testing data, which are prone to several biases. Using data from the REal-time Assessment of Community Transmission-1 (REACT-1) study, a series of cross-sectional surveys assessing prevalence of SARS-CoV-2 infection in England, we estimated the dynamics of England's Omicron wave (from 9 September 2021 to 1 March 2022). We estimate an initial peak in national Omicron prevalence of 6.89% (5.34%, 10.61%) during January 2022, followed by a resurgence in SARS-CoV-2 infections as the more transmissible Omicron sub-lineage, BA.2 replaced BA.1 and BA.1.1. Assuming the emergence of further distinct variants, intermittent epidemics of similar magnitudes may become the 'new normal'.

Since late 2020 SARS-CoV-2 variants of concern (VOCs) have emerged regularly[1–4] leading to substantial changes in national, regional and global dynamics of the COVID-19 pandemic. On 24 November 2021 a new PANGO lineage[5] B.1.1.529 was designated, consisting of genomes sequenced in South Africa and Botswana in the prior week[6], and declared the Omicron VOC by the World Health Organization[7]. Though the Omicron variant has been found to cause less severe disease than previous variants[8,9], it has also been shown to exhibit a large number of

mutations[10] including 15 in the receptor binding domain that has allowed it to escape a majority of pre-existing SARS-CoV-2 neutralising antibodies[11]. Rising incidence in South Africa, following Omicron's emergence, revealed a greater rate of transmission relative to previously dominant VOCs[6]. This has been linked to immune evasion[6,12], including a reduction in the effectiveness of COVID-19 vaccines against Omicron infection[13] and an increased ability to reinfect previously-infected individuals[14]. The increased growth rate has been linked to

[1]School of Public Health, Imperial College London, London, UK. [2]MRC Centre for Global Infectious Disease Analysis and Jameel Institute, Imperial College London, London, UK. [3]Quadram Institute, Norwich, UK. [4]MRC Centre for Environment and Health, School of Public Health, Imperial College London, London, UK. [5]Department of Infectious Disease, Imperial College London, London, UK. [6]Imperial College Healthcare NHS Trust, London, UK. [7]National Institute for Health Research Imperial Biomedical Research Centre, London, UK. [8]Institute of Global Health Innovation, Imperial College London, London, UK. [9]Health Data Research (HDR) UK, Imperial College London, London, UK. [10]UK Dementia Research Institute Centre at Imperial, Imperial College London, London, UK. [11]Department of Statistics, University of Oxford, Oxford, UK. [12]These authors jointly supervised this work: Paul Elliott, Christl A. Donnelly, Marc Chadeau-Hyam. ✉e-mail: o.eales18@imperial.ac.uk; p.elliott@imperial.ac.uk; c.donnelly@imperial.ac.uk; m.chadeau@imperial.ac.uk

both a shorter generation time[15] and a greater number of transmission events per generation[16]. Despite many countries imposing strict travel bans, Omicron rapidly disseminated worldwide, with confirmed cases in 171 countries by 20 January 2022[17].

However, the magnitude of the Omicron wave is not apparent in most countries since testing captures an unknown proportion of infections and is prone to bias due to changing testing capacities and variable/differential test-seeking behaviour[18]. It is likely that in some countries high levels of Omicron infections have saturated testing capacity[19] introducing further bias into estimates of Omicron's dynamics.

Representative community surveys can avoid such biases and accurately measure the prevalence of the virus, with fewer overall tests required[20]. Here, we use data from the REal-time Assessment of Community Transmission-1 (REACT-1) study that has tested randomly selected cross-sections of the population of England approximately monthly since May 2020[21]. We use overall swab-positivity and genomic sequencing from rounds 14 to 18 (9 September 2021 to 1 March 2022) of REACT-1 to describe the dynamics of the Omicron wave in England as it replaced the previously dominant Delta variant. We further explore the diversity of Omicron sub-lineages in round 16 (23 November–14 December), 17 (5 January–20 January) and 18 (8 February–1 March) and how they have contributed to the overall dynamics.

## Results
### Omicron Delta competition
Within the REACT-1 samples we estimated Omicron prevalence of 0.11% (0.07%, 0.16%) by 7 December 2021 (Fig. 1a), three weeks after the first confirmed Omicron case in England was sampled (16 November, linked to recent travel)[22]. At the same date Delta, which had been at a steady high prevalence for the preceding 3 months, was estimated to be approximately twelve-fold higher at 1.31% (1.17%, 1.47%). Though the Omicron variant was likely introduced to England by international travel from Southern Africa[22], we find greater levels of similarity between REACT-1 sequences and sequences sampled in the USA, Germany and France (Supplementary Fig. 1) with inferred high rates of importation/exportation from/to these countries (Supplementary Fig. 2). This likely reflects greater rates of transmission between England and USA/Europe after Omicron was globally disseminated. The proportion of SARS-CoV-2 swab positive cases in England with Omicron rapidly increased, reaching 50% by 14 December (13 December, 16 December) 2021 and 90% by 23 December (20 December, 26 December) 2021 (Fig. 1b). The last Delta sample in REACT-1 (up to round 18) was detected on 14 February 2022 when the proportion of cases linked to Omicron was greater than 99.82%. Our models captured the rapid replacement of Delta with Omicron lineages in England between December 2021 and January 2022.

To further explore the dynamics of the Delta-to-Omicron transition, we estimated a daily growth rate in the log-odds of Omicron infection. The average daily growth rate was estimated at 0.21 (0.20, 0.23) during rounds 16 to 18 (23 November 2021–1 March 2022). Our results suggest that the daily growth advantage varied over time declining steadily from 0.37 (0.28, 0.49) on the 3 December (first day Omicron detected in the REACT-1 study) to 0.11 (0.03, 0.17) on 8 January (Fig. 1c). This change in growth advantage over time could be explained by a shorter generation time for Omicron[15] estimated to be approximately 28% shorter than that of Delta[15]. The decline in growth advantage may also reflect the virus initially achieving higher average rates of transmission among younger, more socially active, and less vaccinated groups than in the population as a whole[23]. A similar decrease in growth advantage over time was detected when the Alpha variant emerged in England during late 2020[3].

Analyses were stratified by region of England (Supplementary Figs. 3–5) and by age (four broad age-groups) (Supplementary

Figs. 6–8). We found similar growth advantages in all regions and age-groups, and a high degree of synchrony in the proportion of Omicron cases across regions. Conversely, we estimated age-specific evolution of the proportion of Omicron cases with an estimated 50% proportion Omicron reached by 10 December (8 December, 14 December) 2021 for 18–34 year olds, while it was reached by 25 December (22 December, 29 December) 2021 in 5–17 year olds. This may reflect age-specific differences in the prevalence of Delta (higher prevalence in 5–17 year olds) before Omicron emerged (Supplementary Fig. 6) and/or age-related differential uptake of vaccination.

As the prevalence of Omicron increased, the prevalence of Delta dropped rapidly to below 0.1% on 3 January 2022 (30 December 2021, 6 January 2022) (Fig. 1a) with similar decreases observed in all regions (Supplementary Fig. 3) and age-groups (Supplementary Fig. 6). Consistently, we estimated that the time-varying reproduction number ($R_t$) for Delta halved in the three weeks from 9 December to 30 December 2021 from 1.00 (0.91, 1.10) to 0.50 (0.38, 0.66) (Fig. 2a). The rapid increase in Omicron infections leading to a depletion of the population susceptible to Delta may at least partially explain this reduction in $R_t$. The contributions of behaviour change[24] and public health measures aimed at reducing transmission[25] to that drop in $R_t$ remains uncertain, though a large decrease in mobility indices for driving, walking and transit were also observed in late-December 2021 (Supplementary Fig. 9).

### Epidemic dynamics of the Omicron wave
Focusing on prevalence of Omicron swab-positivity, we observed a rapid increase with a maximum prevalence of 6.89% (5.34%, 10.61%) reached on 30 December 2021 (21 December 2021, 31 January 2022) (Fig. 1a, Supplementary Fig. 10, Supplementary Table 1). During January, prevalence decreased to 4.18% (4.00%, 4.37%) on 12 January where it plateaued with the beginnings of a resurgence detected in late-January 2022 (Fig. 1a). By early-March 2022 prevalence was approximately constant at 2.60% (2.20%, 3.04%) on 1 March (the last official day of round 18 of the study). Trends in the 2-week average instantaneous reproduction number, $R_t$ (Fig. 2a), showed that on 17 December 2022 (2 weeks after first Omicron detected) $R_t$ was estimated to be 1.99 (1.75, 2.29) despite high levels of vaccine coverage (1 dose 89.8% of those 12 years or older, 2 doses 82.1%, 3 doses 56.9%)[26]. Into early January 2022 $R_t$ rapidly decreased, in line with the sharp decrease in mobility indices over this period (a proxy for social contacts) (Supplementary Fig. 9), with the central estimate falling below one on 2 January 2022 before rising to above one in late January (21 January to 1 February). Through February 2022 $R_t$ was below one, but at the end of round 18 (1 March, 2022), we estimated $R_t$ was no longer securely below one with an $R_t$ value of 1.00 (0.88, 1.12) and 0.50 posterior probability that $R_t > 1$. By 1 March 2022, despite the significantly high levels of recent infections, the herd immunity threshold required for prevalence to decrease had not yet been reached.

During December 2021, Omicron-specific prevalence rapidly increased in all regions of England (Fig. 2b) though there was heterogeneity in the timing and magnitude of the peak (Supplementary Fig. 10, Suppl Table 1). The maximum prevalence reached was highest in the North East at 7.37% (6.42%, 9.79%) and lowest in the East of England at 3.98% (3.38%, 5.80%). Maximum prevalence was reached first in London, peaking at 6.45% (5.15%, 10.27%) on 29 December 2021 (25 December 2021, 28 January 2022), while it was reached on 3 February (11 January, after 1 March) 2022 in South West at 4.12% (3.21%, 6.36%). The rapid rise in prevalence in London could not be explained by a higher regional value of $R_t$ with estimates in December being highly comparable between all regions (Supplementary Fig. 11) and may therefore be related to an earlier introduction of Omicron in London. Trends over time in regional $R_t$ estimates were comparable across regions and followed a similar pattern as the national estimates. On 1 March central estimates of $R_t$ were above one (reflecting

increasing prevalence) in 4 regions: North East, East of England, London and West Midlands.

Trends over time in age-specific $R_t$ also showed a high degree of synchrony (Supplementary Fig. 12). However, $R_t$ in those aged 5–17 years old was higher in January 2022 relative to other age-groups leading to a longer initial period of uninterrupted growth in that age group. $R_t$ for other age-groups dropped below one in early-January 2022. Consistently, the prevalence in 5–17 year olds (Fig. 2c) peaked on 28 January (21 January, 1 February) 2022, reaching a maximum prevalence of 10.74% (8.52%, 14.74%). This was almost 50% higher than the next highest age-group (18–34 year olds) with a maximum

reached 4 weeks earlier on 1 January 2022 (27 December 2021, 5 January 2022) at 7.65% (6.08%, 12.35%) (Supplementary Fig. 10, Supplementary Table 1). The prevalence was lowest in those aged 55 and over with a maximum prevalence of 3.67% (3.25%, 4.88%) reached on 7 January (1 January, after 1 March) 2022., Despite high vaccination rates in those aged 55 years and over[26], there was indication that prevalence in this group (the group most likely to have a severe infections with severe outcomes) was increasing at the end of the study, with an $R_t$ estimate of 1.14 (0.97, 1.33) on 1 March. This demonstrates the limited vaccine effectiveness of COVID-19 vaccines against Omicron infection.

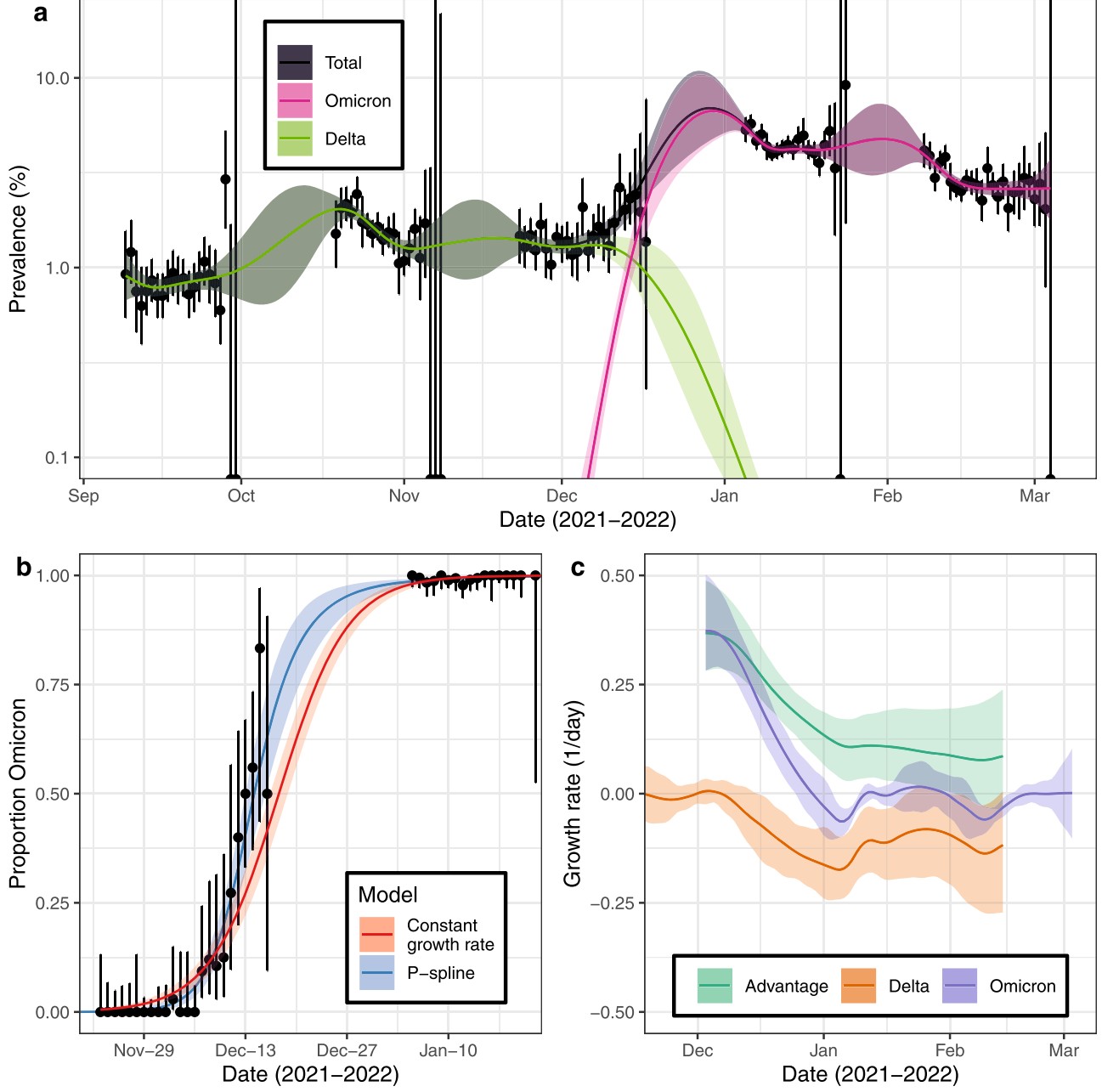

**Fig. 1 | Competition of Omicron and Delta variants. a** Modelled prevalence of SARS-CoV-2 variants Omicron and Delta in England estimated using a mixed-effects Bayesian P-spline model. Estimates of prevalence are shown with a central estimate (solid line) and 95% (shaded region) credible intervals. Daily weighted estimates of mean prevalence (points) are shown with 95% credible intervals (error bars). **b** Modelled proportions of lineages identified as Omicron in England, estimated using Bayesian logistic regression (red) and using a mixed-effects Bayesian P-spline model (blue). Estimates are shown with a central estimate (solid line) and 95% credible intervals (shaded region). Daily estimates of the mean proportion of lineages Omicron (points) are shown with 95% confidence intervals (error bars). **c** Daily growth rate of Omicron (purple), Delta (orange) and their additive difference (green) estimated from the mixed-effects Bayesian P-spline model. Estimates are shown with a central estimate (solid line) and 95% credible intervals (shaded region).

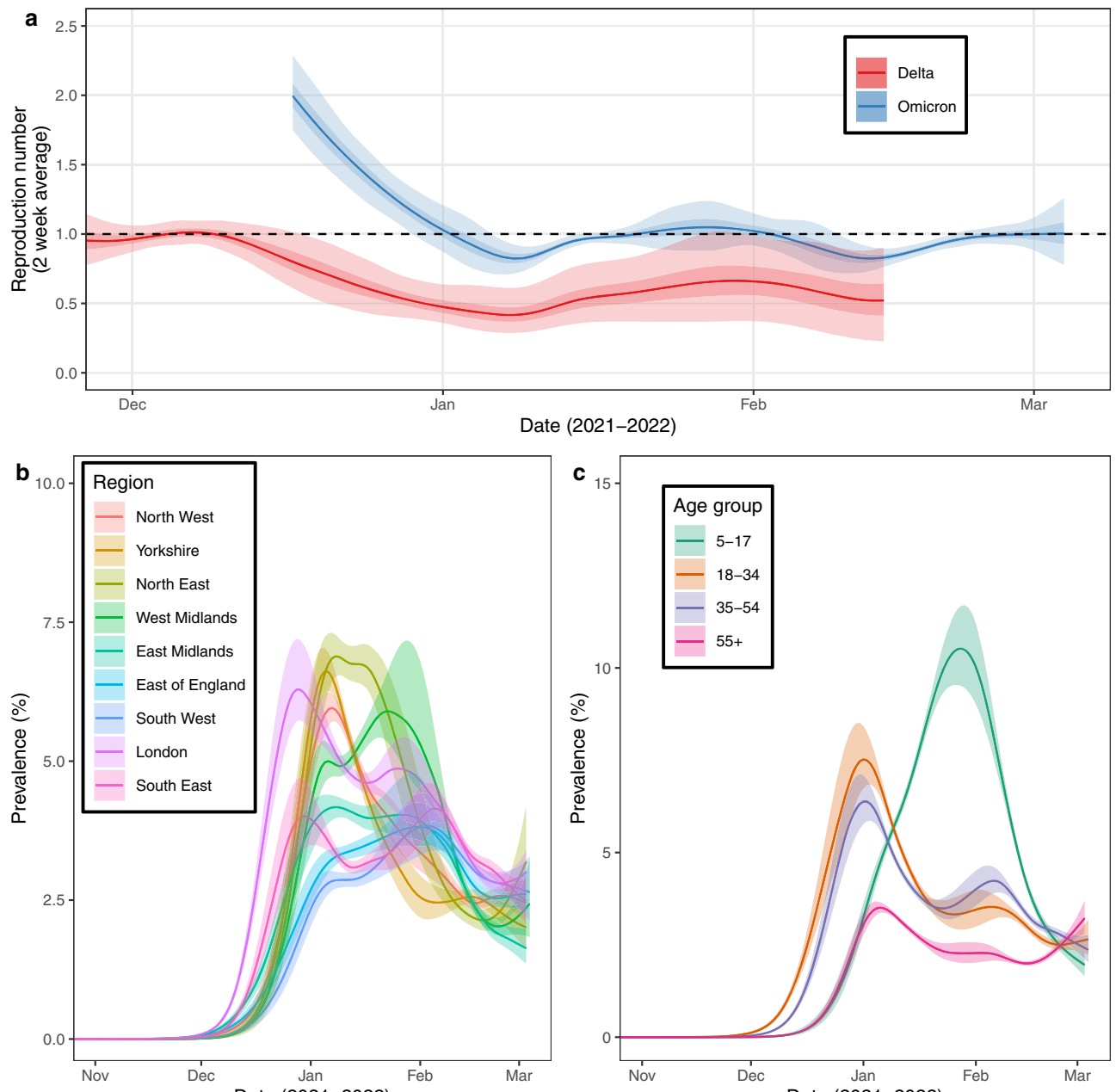

**Fig. 2 | Epidemic dynamics of the Omicron wave. a** Rolling two-week average (prior two weeks) Reproduction number for Omicron and Delta in England as inferred from the mixed-effects Bayesian P-spline model. Estimates are shown with a central estimate (solid line) and 50% (dark shaded region) and 95% (light shaded region) credible intervals. Dashed line shows R = 1 the threshold for epidemic growth. **b** Modelled prevalence of Omicron in each region of England, estimated using a mixed-effects Bayesian P-spline model. Estimates are shown with a central estimate (solid line) and 50% (shaded region) credible intervals. 95% credible intervals and daily point estimates are included in Supplementary figure 1. **c** Modelled prevalence of Omicron for four age-groups in England, estimated using a mixed-effects Bayesian P-spline model. Estimates are shown with a central estimate (solid line) and 50% (shaded region) credible intervals. 95% credible intervals and daily point estimates are included in supplementary figure 4.

## Omicron sub-lineage competition

Multinomial models for the proportion of BA.1, BA.1.1 and BA.2 Omicron sub-lineages showed a decreasing proportion of BA.1 over rounds 16, 17 and 18 of REACT-1 (Supplementary Table 2) with a corresponding increasing proportion of BA.2 increasing over the same period. The proportion of BA.1.1 increased up to 8 February (7 February, 9 February) 2022 and decreased hereafter (Fig. 3a). On 30 December 2021, when Omicron's prevalence reached its maximum, the proportion of BA.1 was at 84.6% (82.9%, 86.2%), BA.1.1 at 15.2% (13.6%, 16.9%) and BA.2 at only 0.2% (0.1%, 0.3%). However, by 1 March, the proportion of BA.1 was 9.6% (8.1%, 11.3%), the proportion of BA.1.1 21.6% (18.7%, 24.9%) and

the proportion of BA.2 was 68.7% (64.6%, 72.7%). Taken together, these results suggest that the winter Omicron wave in England was related to the BA.1 variant and its descendants.

The daily growth rate of the log-odds of BA.1.1 relative to BA.1 was 0.042 (0.037, 0.046), while that of BA.2 relative to BA.1 was 0.133 (0.122, 0.144). The daily growth rate in the log-odds of BA.2 relative to BA.1.1 was 0.091 (0.081, 0.102). This shows that the transmissibility of BA.2 and BA.1.1 are both greater than BA.1, with the highest rate of transmission being for BA.2. This increased growth rate of BA.2 in England has also been detected by the UK Health Security Agency (UKHSA)[27] and multiple other countries have observed an increasing

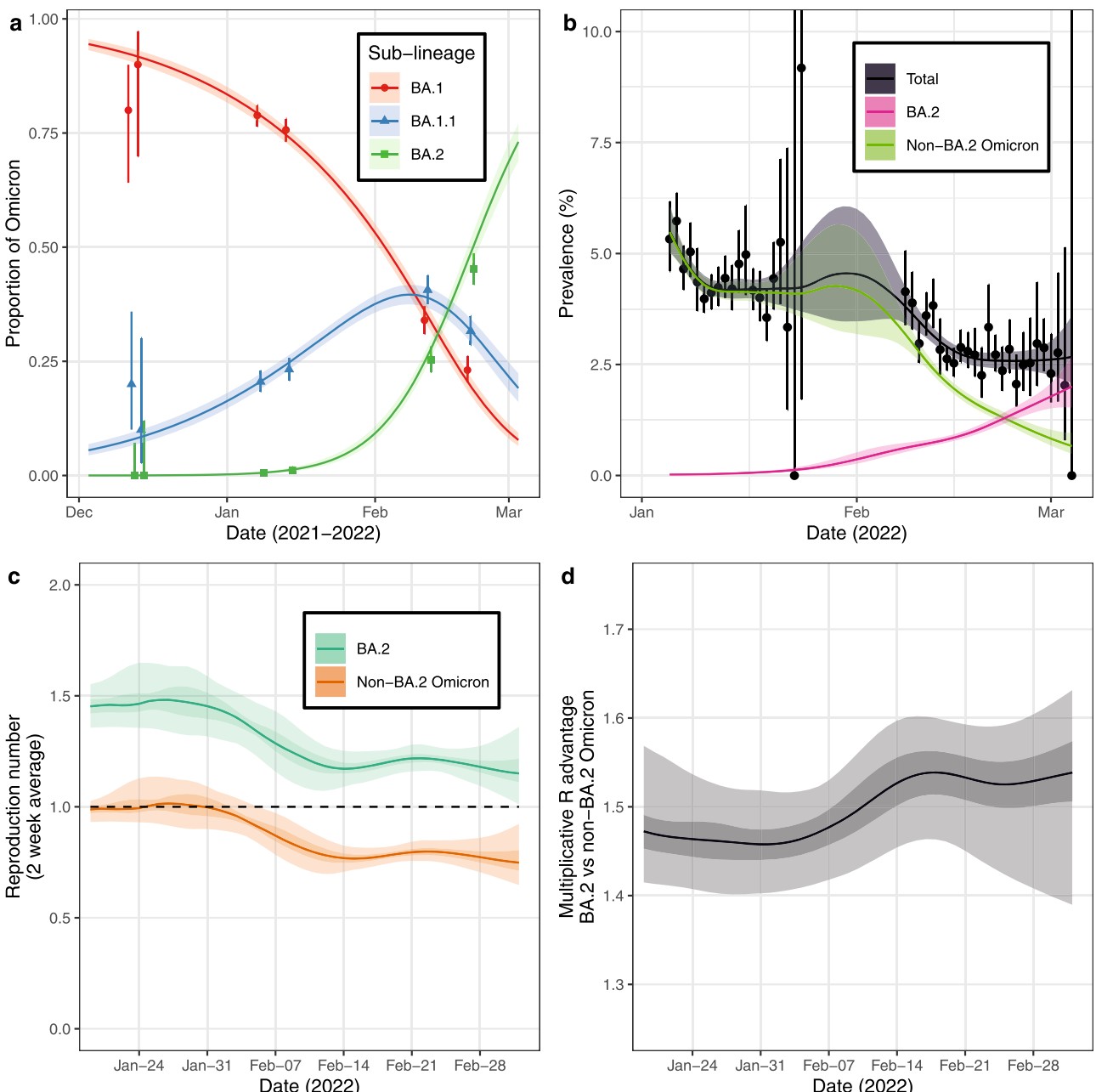

**Fig. 3 | Dynamics of Omicron sub-lineages. a** Modelled proportion of Omicron lineages that are BA.1, BA.1.1 and BA.2, estimated for using a Bayesian multinomial logistic regression model. Estimates of proportion are shown with a central estimate (solid line) and 95% (shaded region) credible intervals. Point estimates of mean proportion (points) are shown with 95% confidence intervals (error bars) for each half round of REACT-1 with x-axis value set by the median date (jittered for all lineages for visualisation). **b** Modelled prevalence of BA.2 and non-BA.2 Omicron in England estimated using a mixed-effects Bayesian P-spline model. Estimates of prevalence are shown with a central estimate (solid line) and 95% (shaded region) credible intervals. Daily weighted estimates of mean prevalence (points) are shown with 95% credible intervals (error bars). **c** Rolling two-week average (prior two weeks) Reproduction number for BA.2 and non-BA.2 Omicron in England as inferred from the mixed-effects Bayesian P-spline model. Estimates are shown with a central estimate (solid line) and 50% (dark shaded region) and 95% (light shaded region) credible intervals. Dashed line shows R = 1 the threshold for epidemic growth. **d** Multiplicative advantage in the two-week average reproduction number for BA.2 vs non-BA.2 Omicron in England as inferred from the mixed-effects Bayesian P-spline model. Estimates are shown with a central estimate (solid line) and 50% (dark shaded region) and 95% (light shaded region) credible intervals.

proportion of BA.2. Increasing proportions of BA.1.1 relative to BA.1 have been observed in Denmark[28], and in the USA BA.1.1 was the major Omicron variant during Omicron's initial emergence[29] suggesting BA.1 was outcompeted before it could establish there. Though most focus has been on the emergence of antigenically distinct variants (BA.2 vs BA.1 for example) there continues to also be a gradual accumulation in beneficial mutations leading to fitter descendant strains (BA.1.1 vs BA.1 for example)

In England, we estimated different trends in the national prevalence of BA.2 vs non-BA.2 Omicron infections over the period of rounds 17 to 18 (Fig. 3b, Supplementary Fig. 13) under the assumption that all infections were Omicron (<1% were non-Omicron infections in both rounds). During February 2022, the prevalence of BA.2 steadily increased, whereas the prevalence of non-BA.2 Omicron decreased. An increasing prevalence of samples that are positive on the S-gene (an approximate proxy for BA.2) was reported in England over the same

period[30]. Consistently, the estimated $R_t$ for BA.2 was greater than that of non-BA.2 Omicron (Fig. 3C) with an estimate of 1.17 (1.08, 1.28) for BA.2 and 0.77 (0.69, 0.87) for non-BA.2 Omicron on 1 March. The difference in $R_t$ over time corresponded to a multiplicative advantage for BA.2 over non-BA.2 Omicron of approximately 1.5 (Fig. 3d), with daily estimates ranging from 1.46 (1.40, 1.52) on 31 January 2022 to 1.54 (1.46, 1.60) on 18 February 2022. This difference in dynamics between Omicron sub-lineages can explain the observed national trends in Omicron prevalence, with $R_t$ increasing towards late February 2022 due to the rising proportion of BA.2.

We found that a greater proportion of BA.2 infected individuals exhibit the most predictive COVID-19 symptoms (loss or change of sense of smell or taste, fever, new persistent cough) 55.3% (51.1%, 59.4%) compared to 45.4% (43.3%, 47.6%) in those infected with BA.1 (Supplementary Tables 3–4). This suggests that symptom-based surveillance and isolation measures could be more effective at identifying BA.2 infected individuals.

Analogous models fit by region (Supplementary Figs. 14–16) and age-group (Supplementary Figs 17–19) showed similar growth rate advantages for BA.2 for all age-groups but a small degree of heterogeneity between region; with a greater advantage in East of England relative to the South West (all other regions were comparable). Although we did not find a higher growth advantage in London and South East, higher proportions of BA.2 were reported in these regions[27]. This may suggest earlier introductions of the Omicron variant in this region, which in-turn, may at least partially be attributed to the higher rates of international travel from these regions[31]. Phylogeographic analysis further supported this with London being highly represented in estimates for the regions of many ancestral nodes within Omicron's phylogenetic tree (Supplementary Fig. 20). However, during periods in which the Omicron sub-lineages were well-sampled there was little geographic structure present in Omicron's phylogeny. This is potentially due to the rapid time-frame by which each Omicron sub-lineage in turn was disseminated across the country following their introduction in London. We estimated higher symmetrical region-to-region migration rates from London to other regions over all rounds and for each Omicron sub-lineage (Supplementary Table 5); they were most consistently high for London to South East and London to North West. Future reactionary measures to the emergence of a new variant would be best targeted at London, and individuals travelling between London and other regions of England.

Analyses by age groups showed that the proportion of BA.2 over time was similar in all age-groups (Supplementary Fig. 18) and therefore could not explain the higher estimates of $R_t$ in those aged 55 and over in late-February/early-March 2022. Central estimates of BA.2-specific $R_t$ were greater than one on 1 March in all regions (Supplementary Fig. 21) and age-groups (Supplementary Fig. 22) and so as the proportion of BA.2 further increases, a resurgence in prevalence would be expected. This has been observed across all regions and age-groups in the numbers of cases and hospitalisations recorded in the routine data during the first three weeks of March 2022[26]. The emergence of BA.2 has acted to prolong the Omicron wave of the epidemic in England.

## Discussion

Here we have presented the dynamics and scale of the Omicron variant wave in England during winter 2021–2022. Most estimates for the magnitudes of different Omicron epidemics worldwide have relied on routine testing data, which are prone to many biases, whereas the REACT-1 data we have used here has fewer biases due to its random sampling procedure. Furthermore, estimates relying on routine testing data often only provide the number of daily positive tests, whereas here we present the prevalence of SARS-CoV-2 infections. This value is fundamentally more important as it reflects the current levels of infection which are directly proportional to the risk of exposure for an individual within the population. Other REACT-1 studies have only considered prevalence of all SARS-CoV-2 infections[32–36], but here we have, using mixed-effects Bayesian P-spline models, estimated daily prevalence of Delta and Omicron SARS-CoV-2 infections separately, and BA.2 and non-BA.2 Omicron infections separately. Previous analysis has also only considered a constant growth advantage between variants[31,33–35,37] whereas as we show here it is likely that the growth advantages between some variants have varied over time. Though the rise of Omicron was rapid, with total prevalence in November 2021 being effectively all Delta, and prevalence in January 2022 being effectively all Omicron, by not treating the observed dynamics as two distinct variants the trends in total prevalence over time risk being overly smoothed when estimated.

Our study has limitations. The sampling is performed over discrete rounds with periods of no data, for which trends have to be inferred. One such period was late December 2021, a key period of growth in the proportion of SARS-CoV-2 infections caused by the Omicron variant, leading to wide credible intervals for the dynamics of the pandemic over this period. Sequencing is unlikely to be successful on samples with a low viral load and so was only performed on samples with an N-gene cycle threshold (Ct) value (a proxy for viral load) less than 34. However, it is unknown if there are intrinsic differences in the Ct values by lineage which could bias estimated proportions. Further, differences in Ct values due to differences in growth rate[38] could also lead to more transmissible variants being detected more favourably.

Finally, our study estimates the daily prevalence, the proportion of the population testing positive, and not the incidence. This can lead to estimates of $R_t$ being overly smoothed due to swab-positivity remaining over several days[39,40], and trends of $R_t$ over time may be lagged by a period depending on the duration for which individuals test positive. It is also worth noting that our estimates of $R_t$ are based on specific estimates of the generation time distribution; studies using different generation time distributions will return different estimates of $R_t$[41]. Additionally, our region- and age-group-specific $R_t$ estimates assume that all infections in a particular subgroup result from contact/mixing among members of that subgroup and so, though highly informative, must be interpreted cautiously.

As the cumulative incidence and vaccination coverage continue to increase, the SARS-CoV-2 virus will find itself competing against a diverse and complex immunity landscape within the human population. Accordingly, the evolutionary dynamics of the virus will be dominated by immune evasion. This has already been observed with the emergence of the Omicron variant and its sub-lineages, the consequence of which was an initial wave of infection peaking at a prevalence of 6.89% in England, the highest recorded at any time hitherto in the REACT-1 study. These infection rates occurred against a background of high levels of vaccine coverage and past infections, further fuelled by the emergence of the more transmissible Omicron sub-lineage BA.2. Given the regular emergence of VOCs during the first two years of the COVID-19 pandemic there is little reason to believe this trend will not continue. Indeed, other respiratory infections such as Influenza observe annual epidemics due to the emergence of new strains better able to navigate the immune landscape[42,43]. If we see a similar trend for SARS-CoV-2 then intermittent waves of infection of a similar magnitude to Omicron are within the bounds of possibility. Continued surveillance, booster vaccinations and, potentially, updates of the vaccines will be crucial in minimising the harmful effects of this new public health paradigm. Greater vaccine equity worldwide can help reduce the rate at which these harmful variants emerge[44].

## Methods

### REACT-1 study protocol

The methodology of REACT-1 has been described in detail elsewhere[45]. In short, each round a random subset of the population in England is selected at the lower tier local authority (LTLA) level ($N = 315$) and

invited to participate in the study. Those who agree to participate provide a self-administered (parent/guardian administered for those aged 5-12 years old) throat and nose swab which undergoes rt-PCR testing for the SARS-CoV-2 virus. Individuals are classified as positive if their test has an N-gene Ct value less than 37 or if both the N- and E-gene are detected. Rim weighting[46] is used to weight individual test results by age, sex, deciles of the Index of Multiple Deprivation, LTLA counts, and ethnic group. Analysis was performed using rounds 14 to 18 of the study running from 9 September 2021 to 1 March 2022. During rounds 15 to 18 of the study all swab tests were sent to the lab via the post, whereas in round 14 of the study approximately 50% of tests were collected via courier. No difference in samples were observed for the two different collection methods[32]. Research ethics approval was obtained from the South Central-Berkshire B Research Ethics Committee (IRAS ID: 283787). All participants who agreed to provide a self-administered test gave informed consent (parent/guardian gave consent for minors).

## Sequencing

All swab tests with an N-gene Ct value less than 34 and sufficient volume underwent genomic sequencing. Extracted RNA was amplified using the ARTIC protocol[47] with sequence libraries provided by CoronaHiT[48]. Sequencing was performed on the Illumina NextSeq 500 platform. Analysis of the raw sequencing was done using the bioinformatics pipeline[49] before being uploaded to CLIMB[50]. Lineage designation was then performed using PangoLEARN[51] (database version 2022-02-28), a machine-learning-based algorithm for lineage designation which uses the PANGO nomenclature[5]. Sequences could not be obtained for some samples of low overall quality. Further, samples for which at least 50% of bases were not covered were excluded from the analysis.

## Mixed-effects Bayesian P-spline model

A mixed-effects Bayesian P-spline model was used to estimate the prevalence of both Omicron and Delta SARS-CoV-2 infections over time. The basic Bayesian P-spline model has been described in detail[52]. In short, the entire time series is split into equally sized knots (approximately 5 days apart) with 3 further knots defined at both the beginning and end of the time series (to prevent edge effects). A system of 4th order basis-splines (b-splines) is defined over all knots. The P-spline for a single lineage's prevalence is then defined as a linear combination of these basis splines:

$$g(\pi(t)) = \sum_i^N b_i B_i(t). \qquad (1)$$

Here $g()$ is the logit link function, $\pi(t)$ is the prevalence on day $t$, $b_i$ are the b-spline coefficients, and $B_i(t)$ is the value of the $i^{th}$ b-spline on day $t$. A second-order random-walk prior distribution is defined for the b-spline coefficients, $b_i = 2b_{i-1} - b_{i-2} + u_i$, where $u_i \sim N(0, \rho)$. The first two coefficients, $b_1$ and, are given uninformative constant prior distributions. The parameter $\rho$ controls the smoothness of the curve and penalises changes in the first derivative (approximately the growth rate) reflecting the prior knowledge we have of an epidemic system. P-splines are defined for the prevalence of both Omicron and Delta with $\rho$ being a shared parameter. A further prior distribution taking the form $u_{i,Omicron} - u_{i,Delta} \sim N(0, \eta)$ is defined on the changes in the first derivative for both lineages. The parameters $\rho$ and $\eta$ were both given uninformative inverse gamma prior distributions $\eta, \rho \sim IG(0.001, 0.001)$. This assumes that changes in the growth rate happen simultaneously for both Omicron and Delta, which effectively assumes a constant growth rate advantage, unless there is significant evidence to the contrary.

The sum total of Omicron and Delta's modelled prevalence is fitted to the daily weighted number of tests and positive tests assuming a binomial likelihood. Simultaneously the proportion of the total prevalence attributed to Omicron is fitted to the daily number of Omicron lineages vs total number of samples with a lineage determined, again assuming a binomial likelihood. The model is fit to rounds 14 to 18 of the REACT-1 data using a No-U-Turns sampler[53] implemented in STAN[54]. Models are also fit to the data subset by region of England, and subset by age quartile. An analogous model is fit to rounds 17 and 18 instead comparing BA.2 and non-BA.2 Omicron lineages under the assumption that total prevalence was caused only by Omicron lineages (>99% Omicron in both rounds).

Estimates of the instantaneous growth rate of Omicron, Delta and their difference were estimated over time from the modelled prevalence time-series. The rolling two-week average reproduction number for both Omicron and Delta was also estimated from the modelled prevalence using methodology that has previously been described[52]. The model assumed a gamma-distributed generation time with rate parameter = 0.27, and shape parameter = 0.89, for Omicron or Omicron sub-lineages (BA.2, non-BA.2 Omicron lineages) and rate parameter = 0.48 and shape parameter = 2.20 for Delta[15].

## Constant growth rate models

The daily growth rate in the log-odds of Omicron infection relative to Delta infection, assuming a constant growth rate for the whole period, was estimated using a Bayesian logistic regression model fit to a binary outcome variable (Omicron or Delta) over time, implemented using the brms R package[55]. The daily growth rate in the log-odds of BA.2 and BA.1.1 relative to BA.1 over rounds 17 and 18, assuming constant growth rates, was estimated using a Bayesian multinomial logistic regression model fit to the categorical outcome variable (BA.1, BA.1.1 or BA.2) with BA.1 set as the reference category. The difference in these two growth rates was used to estimate the daily growth rate in the log-odds of BA.2 relative to BA.1.1.

## Phylogeographic analysis

Phylogeographic analysis was performed on lineages that were designated as Omicron (BA.1, BA.1.1, BA.2). A maximum likelihood phylogenetic tree assuming a HKY model was fitted to the sequences using IQTree[56]. A relaxed molecular clock model, assuming a mean evolutionary rate of 0.0008 substitutions/site/year, was fit to the tree using TreeTime[57] to give a time-resolved phylogenetic tree. A further mugenic model again implemented in TreeTime[57] was fit to the time-resolved phylogeny treating the region of England (N = 9) where each sample was obtained as a discrete state. From this model we estimated the mean pairwise migration rate of Omicron between all regions of England. We excluded sequences without complete date information since they do not contribute to the estimation of divergence times. Sequences with an excess of gaps cannot be placed in the phylogeny correctly, and so we excluded sequences with less than 75% of bases covered. Note that this is a more stringent threshold than was used earlier for the task of lineage classification which can be performed for sequences with fewer bases covered. We further excluded one sequence which deviated too much from a preliminary strict clock (more than five times the interquartile range from the clock regression).

Omicron sequences with at least 75% of bases covered were compared to all sequences deposited in GISAID after the 2nd Dec 2021, with the 500 closest neighbours extracted using uvaiann[58]. The similarity measure is based on the number of single nucleotide polymorphism (SNP) matches, number of partial matches, and number of valid comparisons so that we prefer more resolved sequences. Matches to GISAID samples from the REACT-1 study were excluded afterwards. We then compared the number of samples that match (no different SNPs) a REACT-1 sequence by the country where they were collected. We additionally investigated the number of samples that were at specific SNP distance (1,2,3 and 4 SNP mismatches) from REACT-1 sequences again by the country where they were collected.

The reported number of SNP mismatches considers partially ambiguous DNA codes: DNA bases which cannot be unambiguously inferred by the assembler may be reported as e.g., 'M' to indicate that the base may be an adenine (A) or a cytosine (C)[59]. Such a state is compatible and thus considered a match to another sequence which has for instance an 'A' in the same genomic location.

In order to estimate potential Omicron importations into and exportations from England based on the REACT-1 samples, we used the set of closest neighbours described above, restricting to global sequences sampled within one week of REACT-1 samples with at most one SNP distance. We furthermore removed global GISAID matches from the UK (i.e., sequences deposited in GISAID from the UK), and for each REACT-1 sequence we kept only the global match with the highest number of valid pair comparisons (locations where neither sequence is a gap or have low coverage). After removing duplicate hits, since the same global reference can be the best match for more than one REACT-1 sequence, we inferred a potential importation if the sampling date of the global sequence is earlier than its matching REACT-1 sample, and as a potential exportation if the REACT-1 sample is earlier (but still within one week). Global sequences which matched both an earlier and a later REACT-1 sample were removed (since they could be inferred as a source or destination of the migration). In total, we have 335 imported samples and 310 exported ones. The date of the importation/exportation was taken as the date of the second sample for each pair. The actual import/export dates may be earlier due to an importation lag[60].

### Statistical analyses

The Wilson method[61] which is preferred for low numbers of positives[62] was used to calculate the 95% confidence intervals for all lineage proportions.

The proportion of individuals reporting any symptoms, and the proportion reporting the most predictive COVID-19 symptoms (loss or change of sense of smell or taste, fever, new persistent cough)[63] in the last month was estimated in round 16 for those infected with Omicron and Delta, and in rounds 17–18 for those infected with BA.1, BA.1.1 and BA.2. The combination of lineages and rounds chosen was done to avoid introducing biases due to changing rates of symptoms over time, and to ensure a large enough sample of each lineage for calculations to be meaningful. $P$-values for differences in the proportion reporting symptoms between lineages was estimated using logistic regression models with symptom status (any symptom vs no symptoms and separately most predictive COVID-19 symptoms vs not reporting the most predictive COVID-19 symptoms) as the outcome variable. The sensitivity of any result that was significant ($P$-value <0.05) was assessed using multivariable logistic regression models including round of the study and N-gene Ct value as additional covariates. Statistical analyses were performed using R software, version 4.0.5.

### Apple mobility data.

Daily data for mobility indices in England for driving, walking and transit were downloaded from Apple mobility trend reports[58]. Seven-day moving averages were estimated from the daily data and scaled so that the maximum value over the period of 1 December 2021 to 1 March 2022 was 100.

### Reporting summary

Further information on research design is available in the Nature Research Reporting Summary linked to this article.

## Data availability

Access to REACT-1 individual-level data is restricted to protect participants' anonymity.

Summary statistics and data, descriptive tables, and code including the daily weighted number of tests, weighted number of positive tests and daily number of Delta, BA.1, BA.1.1 and BA.2 samples (used for the P-spline models) from the current REACT-1 study are available at https://github.com/mrc-ide/reactidd (https://doi.org/10.5281/zenodo.6557251).

Sequence read data are available without restriction from the European Nucleotide Archive at https://www.ebi.ac.uk/ena/browser/view/PRJEB37886, and consensus genome sequences are available from the Global initiative on sharing all influenza data (GISAID). The accession numbers are provided in the supplementary data (Supplementary Data 1).

Requests for materials should be made to Paul Elliott, p.elliott@imperial.ac.uk, School of Public Health, Imperial College London, Norfolk Place, London, W2 1PG. Aggregate data can only be shared if there is an appropriate number of individuals in each category such that data remains unidentifiable. For more information on the questions that are asked to participants (data variables available) please refer to the REACT-1 study materials (https://www.imperial.ac.uk/medicine/research-and-impact/groups/react-study/for-researchers/react-1-study-materials/). A response to requests should normally be received within a month of the request being made.

## Code availability

Code is available at https://github.com/mrc-ide/reactidd[64]

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

## Acknowledgements

The study was funded by the Department of Health and Social Care in England. The funders had no role in the design and conduct of the study; collection, management, analysis, and interpretation of the data; and preparation, review, or approval of this manuscript.

A.J.P. acknowledges the support of the Biotechnology and Biological Sciences Research Council (BB/R012504/1). H.Ward acknowledges support from a National Institute for Health Research (NIHR) Senior Investigator Award, the Wellcome Trust (205456/Z/16/Z), and the NIHR Applied Research Collaboration (ARC) North West London. G.C. is supported by an NIHR Professorship. P.E. is the Director of the Medical Research Council (MRC) Centre for Environment and Health (MR/L01341X/1, MR/S019669/1). P.E. acknowledges support from Health Data Research UK (HDR UK); the NIHR Imperial Biomedical Research Centre; NIHR Health Protection Research Units in Chemical and Radiation Threats and Hazards, and Environmental Exposures and Health; the British Heart Foundation Centre for Research Excellence at Imperial College London (RE/18/4/34215); and the UK Dementia Research Institute at Imperial College London (MC_PC_17114). S.R. and C.A.D. acknowledge support from the MRC Centre for Global Infectious Disease Analysis. C.A.D. acknowledges support from the NIHR Health Protection Research Unit in Emerging and Zoonotic Infections and the NIHR-funded Vaccine Efficacy Evaluation for Priority Emerging Diseases (PR-OD-1017-20007). M.C.-H. and B.B. acknowledge support from Cancer Research UK, Population Research Committee Project grant 'Mechanomics' (Grant No. 22184 to MC-H). MC-H acknowledges support from the H2020-EXPANSE (Horizon 2020 grant No. 874627) and H2020-LongITools (Horizon 2020 grant No 874739).

We thank The Huo Family Foundation for their support of our work on COVID-19. We thank key collaborators on this work—Ipsos MORI: Kelly Beaver, Sam Clemens, Gary Welch, Nicholas Gilby, Kelly Ward, Galini Pantelidou and Kevin Pickering; Institute of Global Health Innovation at Imperial College London: Gianluca Fontana, Justine Alford; School of Public Health, Imperial College London: Eric Johnson, Rob Elliott, Graham Blakoe; Quadram Institute, Norwich, UK: Nabil-Fareed Alikhan; North West London Pathology and Public Health England (now UKHSA) for help in calibration of the laboratory analyses; Patient Experience Research Centre at Imperial College London and the REACT Public Advisory Panel; NHS Digital for access to the NHS register; the Department of Health and Social Care for logistic support.

We thank GISAID for providing global SARS-CoV-2 sequences and gratefully acknowledge all authors (Supplementary Data 2) from the originating laboratories responsible for obtaining the specimens and the submitting laboratories where genetic sequence data were generated and shared via the GISAID Initiative.

## Author contributions

P.E., C.A.D., M.C.-H. and O.E. are corresponding authors. O.E., S.R., P.E., M.C.-H. and C.A.D. conceived the study and the analytical plan. O.E., L.O.M., A.P. performed the statistical analyses. O.E., L.O.M., A.P., H.W., B.B., D.T., D.H., and J.J. curated the data. C.A., D.A., W.B., G.T., G.C., H.Ward, A.D. provided study oversight and results interpretation. All authors revised the manuscript for important intellectual content and approved the submission of the manuscript. O.E., L.O.M., A.P., P.E., M.C.-H., C.A.D. had full access to the data and take responsibility for the integrity of the data and the accuracy of the data analysis and for the decision to submit for publication.

## Competing interests

A.D. is chairman of the Health Security and Pre-Emptive Medicine Initiative, Flagship Pioneering UK plc, and has no conflict of interest to declare. M.C.-H. holds shares in the O-SMOSE company. Consulting activities conducted by the company are independent of the present work, and M.C.-H. has no conflict of interest to declare. All other authors have no competing interests to declare.
