## [Peer Review File · Nature Communications]

Dynamics of competing SARS-CoV-2 variants during the Omicron epidemic in EnglandREVIEWER COMMENTS

Reviewer #1 (Remarks to the Author):

The authors provide a very detailed analysis of the Omicron transmission dynamics in the England, four months after its global emergence. The conclusions of the manuscript are well supported by the analysis and limitations are acknowledged.

I only have one question or suggestion for the authors. Could they provide a figure and some discussion of how changes in government restrictions affected the observed sublineage dynamics e.g. the R_t during the study period?

Reviewer #2 (Remarks to the Author):

This study by Eales et al. presents unbiased prevalence estimates from the REACT-1 study in England. This is incredibly valuable data, as it is difficult to obtain these estimates from routine testing, especially in an era where testing becomes less and less common.

This reviewer has two main concerns: (1) Although important, the data presented doesn't give us that many novel insights, and (2) The manuscript is very descriptive, which makes it rather uninteresting to read. The manuscript reads as one big sausage of words, and at the end of the manuscript it is hard to figure out what the key finding/message of the paper is. This could be considerably improved by ending each paragraph with a conclusion of what we have learned from the data. For example, paragraph L124-136 ends with the following sentence: "On 1 March central estimates of R_t were above one in 4 regions: North East, East of England, London and West Midlands." What does this mean? Why is this important? The majority of the paragraphs could be substantially improved by answering these two questions.

Reviewer #3 (Remarks to the Author):

Omicron has been shown to be less severe, but has been characterised by immune evasion either through vaccine escape or reinfection. The authors use data from the REACT-1 project (a survey of PCR testing and genomic sequencing) and a mixed-effects Bayesian P-spline model was used to estimate peaks of Omicron prevalence during winter and spring 2022. Using data from the same period - the authors estimated an initial peak in national Omicron prevalence of 6.89% (5.34%, 10.61%) during January 2022, followed by a resurgence in SARS-CoV-2 infections in England during February-March 2022 during a more transmissible Omicron sub-lineage BA.2 wave.

The work is of significance as these estimated peaks in prevalence are important to understand - these studies use a sound reproducible methodology to provide a unique insight into the dynamics of the COVID-19 pandemic during the periods where new variants of concern and sub-lineages emerge. These data also help with forecasting future waves of COVID-19. The work supports the study conclusions and there are no apparent flaws in the data, analysis, interpretation and conclusions.

Minor:

The Abstract section could be improved with reference made to the database used, the study design/main method for making their estimates - more information describing declining prevalence and plateau during the BA.2 wave in Feb/March 2022 should also be included. It should also be clarified in the Abstract that this is a retrospective study design.

Omicron prevalence peaks had regional and age variations and it is perhaps worth also providing a highlight of these results in the abstract.

REVIEWER COMMENTS

Reviewer #1 (Remarks to the Author):

The authors provide a very detailed analysis of the Omicron transmission dynamics in the England, four months after its global emergence. The conclusions of the manuscript are well supported by the analysis and limitations are acknowledged.

I only have one question or suggestion for the authors. Could they provide a figure and some discussion of how changes in government restrictions affected the observed sublineage dynamics e.g. the R_t during the study period?

We have added an additional supplementary figure showing mobility indices over the same period that we have shown R_t estimates for.

We have extended the sentence discussing the decrease in Delta's R_t to "The contributions of behaviour change [24] and public health measures aimed at reducing transmission [25] to that drop in R_t remains uncertain, though a large decrease in mobility indices for driving, walking and transit were also observed in late-December 2021 (Sup Fig. 9)."

We have also extended the discussion of Omicron's R_t to "In early January 2022 R_t rapidly decreased, in line with the sharp decrease in mobility indices over this period (a proxy for social contacts) (Supplementary Fig. 9)..."

Reviewer #2 (Remarks to the Author):

This study by Eales et al. presents unbiased prevalence estimates from the REACT-1 study in England. This is incredibly valuable data, as it is difficult to obtain these estimates from routine testing, especially in an era where testing becomes less and less common.

This reviewer has two main concerns: (1) Although important, the data presented doesn't give us that many novel insights, and (2) The manuscript is very descriptive, which makes it rather uninteresting to read. The manuscript reads as one big sausage of words, and at the end of the manuscript it is hard to figure out what the key finding/message of the paper is. This could be considerably improved by ending each paragraph with a conclusion of what we have learned from the data. For example, paragraph L124-136 ends with the following sentence: "On 1 March central estimates of R_t were above one in 4 regions: North East, East of England, London and West Midlands." What does this mean? Why is this important? The majority of the paragraphs could be substantially improved by answering these two questions.

We have now included interpretative statements throughout the results ensuring that the section does not read too descriptively. In the quoted sentence, the phrase "(reflecting increasing prevalence)" was added for greater clarity.

Reviewer #3 (Remarks to the Author):

Omicron has been shown to be less severe, but has been characterised by immune evasion either through vaccine escape or reinfection. The authors use data from the REACT-1 project (a survey of PCR testing and genomic sequencing) and a mixed-effects Bayesian P-spline model was used to estimate peaks of Omicron prevalence during winter and spring 2022. Using data from the same period - the authors estimated an initial peak in national Omicron prevalence of 6.89% (5.34%, 10.61%) during January 2022, followed by a resurgence in SARS-CoV-2 infections in England during February-March 2022 during a more transmissible Omicron sub-lineage BA.2 wave.

The work is of significance as these estimated peaks in prevalence are important to understand - these studies use a sound reproducible methodology to provide a unique insight into the dynamics of the COVID-19 pandemic during the periods where new variants of concern and sub-lineages emerge. These data also help with forecasting future waves of COVID-19. The work supports the study conclusions and there are no apparent flaws in the data, analysis, interpretation and conclusions.

We thank the reviewer for this positive feedback.

Minor:

The Abstract section could be improved with reference made to the database used, the study design/main method for making their estimates - more information describing declining prevalence and plateau during the BA.2 wave in Feb/March 2022 should also be included. It should also be clarified in the Abstract that this is a retrospective study design.

We have clarified in the abstract that the study design was a series of cross-sectional surveys and that there were two Omicron waves (the first BA.1 based and the second BA.2 based).

Omicron prevalence peaks had regional and age variations and it is perhaps worth also providing a highlight of these results in the abstract.

We have now mentioned the age and regional analyses we conducted in the abstract.